# A Review of Supercapacitors Based on Graphene and Redox-Active Organic Materials

**DOI:** 10.3390/ma12050703

**Published:** 2019-02-27

**Authors:** Qi Li, Michael Horn, Yinong Wang, Jennifer MacLeod, Nunzio Motta, Jinzhang Liu

**Affiliations:** 1School of Materials Science and Engineering, Beihang University, Beijing 100083, China; qi_li@buaa.edu.cn; 2School of Chemistry, Physics, and Mechanical Engineering, Queensland University of Technology, Brisbane, QLD 4001, Australia; michael.horn@hdr.qut.edu.au (M.H.); jennifer.macleod@qut.edu.au (J.M.); n.motta@qut.edu.au (N.M.); 3School of Mathematics and Physics Science, Dalian University of Technology, Panjin 124221, Liaoning, China

**Keywords:** graphene, conducting polymers, supercapacitors, organic molecules, flexible devices

## Abstract

Supercapacitors are a highly promising class of energy storage devices due to their high power density and long life cycle. Conducting polymers (CPs) and organic molecules are potential candidates for improving supercapacitor electrodes due to their low cost, large specific pseudocapacitance and facile synthesis methods. Graphene, with its unique two-dimensional structure, shows high electrical conductivity, large specific surface area and outstanding mechanical properties, which makes it an excellent material for lithium ion batteries, fuel cells and supercapacitors. The combination of CPs and graphene as electrode material is expected to boost the properties of supercapacitors. In this review, we summarize recent reports on three different CP/graphene composites as electrode materials for supercapacitors, discussing synthesis and electrochemical performance. Novel flexible and wearable devices based on CP/graphene composites are introduced and discussed, with an eye to recent developments and challenges for future research directions.

## 1. Introduction

The continued use of fossil fuels, combined with limited reserves, has resulted in two serious and urgent issues to be addressed in our daily lives; environmental pollution and concern around continued energy supply. To address these problems, scientists focus on looking for clean and renewable energy sources such as solar energy and wind energy. However, these renewable energy sources are difficult to store, meaning that the development of novel, efficient, and low cost energy storage technology is very significant. In recent years, fuel cells, ionic batteries and electrochemical supercapacitors have been investigated [1,2,3]. Compared with batteries, supercapacitors possess the advantages of fast charge/discharge (high power) and long life cycle, making them appealing for a wide range of applications in energy storage, such as electric vehicles and remotely deployed equipment [4,5,6].

Based on the different mechanisms of energy storage, supercapacitors can be broadly classified into two types: electric double layer capacitors (EDLCs) and pseudocapacitive enhanced devices, sometimes referred to as pseudocapacitors [7,8,9,10]. EDLCs store charge by electrostatic interactions, promoting physical adsorption and desorption of ions at the electrolyte/electrode interface. In other words, during the process of charge and discharge the electrode materials do not partake in a chemical reaction. Transport of ions, followed by adsorption/desorption, can occur within seconds granting to EDLCs fast charge and discharge capabilities. The specific capacitance of EDLCs is affected by surface area, pore structure and pore size distribution [11,12,13]. As a result, carbon based materials, such as activated carbon, carbon nanotubes and graphene are often used [14,15,16].

For pseudocapacitors, the energy storage mechanism is based on fast and reversible Faradaic redox reactions between an electrolyte and electrode. In general, the specific capacitance of pseudocapacitors is higher than that of EDLCs based on carbon materials, because the reaction occurs both on the surface and in the bulk of electrode materials. As a drawback to this, volume expansion and contraction of the materials during the charge/discharge process results in lower cyclic stability [17,18]. Conducting polymers (CPs), such as polyaniline (PANI), polypyrrole (PPy), and metal oxides (MnO_2_, NiO_x_, etc.) are typical active materials for pseudocapacitors [19,20,21,22]. The charge stored in devices described as pseudocapacitors is generally a combination of both pseudocapacitance, and the electric double layer (EDL) capacitance contributed by the surface of the active material.

Over the past few years, many carbonaceous nanomaterials were used for supercapacitor electrodes. Single-walled carbon nanotube (SWCNT) is one of commonly used carbon nanomaterials, with a theoretical specific surface area of 1300 m^2^/g and also exhibit good specific capacitance as supercpacitor electrodes [23]. However, the limited surface area of SWCNT and high production cost restrict their applications for supercapacitor [24]. In comparison, graphene is a promising two-dimensional (2D) material for supercapacitors, with a theoretical specific surface area of 2630 m^2^/g [25]. Furthermore, graphene also shows high electrical conductivity, wide potential window, good chemical stability, and outstanding flexibility [25,26,27]. Graphene synthesized by chemical vapor deposition (CVD) and mechanical exfoliation exhibits excellent properties, but due to the high manufacture cost and low production efficiency, graphene synthesized by these methods is unsuitable for supercapacitors. In contrast, graphene oxide (GO) and reduced graphene oxide (rGO), which are fabricated via wet chemical methods, is cheap and convenient. The EDL capacitance of single layer graphene has been measured by researchers and their results reveal that the theoretical capacitance of graphene is up to ~550 F/g, when the full surface area of the graphene is able to be utilized [25,28]. However, when processed into electrode films via various methods, graphene layers tend to stack and aggregate inhibiting electrolyte access to a great part of the surface area. As a result, supercapacitors based on graphene show a low specific capacitance, about 200 F/g [29].

In recent years, many pseudocapacitive materials have been investigated and used to fabricate supercapacitor electrodes in combination with rGO. These include MnO_2_, NiCo_2_O_4_, Ni(OH)_2_, and CPs [30,31,32,33,34]. These composite supercapacitor electrodes generally show excellent capacitance. This is believed to be because the included compounds provide pseudocapacitance, which comes in addition to the EDL capacitance provided by high surface area rGO. In addition, compared to pure pseudocapacitive materials, the composites have a long life cycle. Thus, these materials attract a lot of attention in supercapacitor applications. Compared with transition metal oxides or hydroxides, CPs are considered as a potential candidate for supercapacitors due to their low cost, good electrical conductivity, and large theoretical specific capacitance [18,35,36]. PANI, PPy and poly(3,4-ethylenedioxythiophene) (PEDOT) are three main CPs which are often investigated and used to fabricate supercapacitor electrodes. However, CP-only electrodes exhibit poor cyclic stability because of volume changes during charge/discharge processes, which limits their wider application. A large number of studies demonstrate that graphene and CP composites, when used for electrode materials of supercapacitors, exhibit outstanding electrochemical and mechanical behaviors [37,38,39]. On this basis, researchers found that some organic molecules could be used to improve the properties of graphene-based supercapacitors.

In this review, we summarize recent research progress in graphene/CP composites for application to supercapacitors. The content is divided into five sections; following the introduction of graphene and CPs applied to the field of supercapacitors, Section 2 will discuss graphene/CP composites used for electrodes. In Section 3, the benefits of combining graphene with organic molecule doping is explored. Section 4 covers novel supercapacitor architectures (flexible supercapacitors and wearable supercapacitors) based on graphene/CPs electrodes. Finally, in Section 5, we summarize the current limitations of supercapacitors based on graphene/CPs composites, and discuss possible research directions for the future.

## 2. Graphene/CPs Composites 

CPs consist of chains of single (C–C) and double (C=C) carbon bonds, which are electrically conducting through a conjugated bond backbone. It is easy to synthesize them by chemical oxidation (for example with iron chloride as the oxidant), or via electrochemical oxidation. In general, CPs with high theoretical specific capacitances and high charge mobilities are considered as good pseudocapacitive materials for building supercapacitors with higher energy densities, compared to the conventional carbon-based EDLCs. However, the stability of CPs is very poor when exposed to extended cycling. During charge/discharge processes, repeated insertion/de-insertion of ions causes material volumetric changes, leading to swelling, shrinkage, and even cracking [40]. As a result, the specific capacitance drops rapidly, failing to meet the requirements of application. To solve the problem, researchers have tried combining graphene with CPs. The network structure of rGO successfully improves the stability and capacitance retention of CPs. This section will further discuss some examples of graphene/CP composites and their application as supercapacitor electrodes.

### 2.1. Graphene and PANI Composites

PANI is a promising candidate for supercapacitor electrodes due to its low cost, good stability, high pseudocapacitance, and easy synthesis. PANI needs protons to be properly charged and discharged and, therefore, it requires a protic solvent, an acidic solution or a protic ionic liquid as the electrolyte, when the material is used for energy storage [18,41]. To overcome the poor mechanical cycle stability of PANI, graphene may be used as a supporting scaffold. The graphitic material also provides a good environment for PANI growth during synthesis. Nanostructured PANI/graphene composites can be synthesized via different methods such as templating [42,43,44], electrochemical polymerization [43,44], and chemical polymerization techniques [45,46].

In one of the earliest known reports on flexible composite films from chemically converted graphene (CCG) and PANI nanofibers (PANI-NFs), Wu et al. used an easy vacuum filtration method for production [47]. Due to the flow-assembly effect of graphene, the composite films (G-PNF_30_) had a layered structure where PANI-NFs was found to be sandwiched between CCG layers (Figure 1a,b). The resulting film had a specific surface area of 12.7 m^2^ g^−1^. The typical cyclic voltammetry (CV) of the composite film, used as an electrode, exhibited a somewhat rectangular area with two redox peaks, attributed to the change in oxidation states of PANI (Figure 1c). For comparison, the CV of untreated CCG shows the true quasi-rectangular area characteristic of double layer capacitance. The composite films can be seen to possess both the characteristics of CCG and that of stand-alone PANI-NFs, both also shown in Figure 1c. In addition, Figure 1d shows the comparative “IR drop”, suggesting that the graphene layers enhanced the electrical conductivity of PANI-NFs. Furthermore, the cyclic stability of pure PANI-NFs was poor, losing 29% after 800 cycles at 3 A g^−1^, while the capacitance of G-PNF_30_ film decreased only 21% under the same conditions. Finally, a symmetric supercapacitor device fabricated from G-PNF_30_ film achieved a specific capacitance of 210 F g^−1^ at 0.3 A g^−1^, which is not high compared to the theoretical capacitance of PANI (~2000 F g^−1^).

In most cases, electrochemical or chemical polymerization are the preferred methods to synthesize graphene/CPs composites. Electrochemical polymerization only requires dissolving the monomer in acid, and subsequently the PANI monomer can be polymerized by applying a suitable electric potential. The electrochemical method provides flexibility in the polymerization reaction by adjusting the range of current, scan rate, and scan direction. Wang et al. combined flow-directed assembly and in situ anodic electrochemical polymerization techniques to prepare a graphene/PANI composite paper (GPCP) with good capacitive performance [44,48]. The GPCP-900s showed a peak capacitance of 233 F g^−1^ and volumetric capacitance of 135 F cm^−3^. Li et al. also synthesized electrodes based on graphene nanosheet/PANI (GNS/PANI) composites using in situ polymerization, which exhibited a remarkable specific capacitance of 1130 F g^−1^ at a scan rate of 5 mV s^−1^ in 1 M H_2_SO_4_ solution, with a capacitance retention of 87% after 1000 cycles [46]. When using these two methods, aniline monomers tend to adsorb on the surface of rGO homogeneously first and then polymerize, resulting in a PANI film coated on individual 2D graphene sheets. The homogeneous films protect the graphene from aggregation during the reaction. This unique network structure effectively provides a large surface area which improves the ion and electron transfer.

In spite of the ease of the above synthesis methods, it is difficult to control the morphology and distribution of PANI-based materials in chemical and electrochemical polymerization. In addition, when using porous rGO hydrogels, sometimes non-uniform molecular-level PANI may block the pores and hinder the electrolyte transfer [49]. Recently, many other methods were used to synthesize the graphene/PANI composites. Wu et al. designed a new self-assembly method for preparing a 3D porous PANI/rGO composite gel (PGG) [48]. As shown in Figure 2a, the first step is the assembly of PANI on GO sheets performed in a water/N-methyl-2-pyrrolidone (NMP)-blended solvent. After that, the obtained PANI/GO composite sheets are reduced and, during the process, self-assemble into a porous 3D material. The rGO sheets covered by PANI molecules form a porous network suitable for the ion transport required by an electrode. When tested, this material achieved a high specific capacitance of 808 F g^−1^ (5157 mF cm^−2^) at a high current density of 53.33 A g^−1^ (377.4 mA cm^−2^). As illustrated in Figure 2c, the reason for the excellent rate performance is that, as PANI is incorporated with GO prior to the reduction of GO, PANI is already in place as the GO nanosheets self-assemble and form a 3D gel structure. This means there is less likelihood that the PANI will block ion transport channels, as it may if were coated onto an already assembled rGO gel (Figure 2b). In addition, the authors investigated the influence of the proportion of PANI content, as well as the PANI solvent type, for their influence over precisely controlling the microstructure. When the content of PANI was too high, it was found that the PANI coating got thicker and the rGO sheets became curled, blocking the electrolyte transfer. When PANI was put into a solvent in which it was less soluble (e.g., DMSO), some PANI chains will nucleate, yielding large aggregates. It was found that the larger the PANI particles are, the faster the capacity drops with the current density. The investigation demonstrates a promising method for synthesizing PANI/rGO based composite electrodes. 

Further improvements in the electrochemical performance of graphene/PANI composites can be achieved by controlling the structure and enhancing the surface area. Zhang et al. proved that different GO concentrations lead to different morphologies of the GO/PANI composite with different electrochemical behaviors [50]. Li et al. prepared graphene/PANI nanocomposites by reducing GO with hydrazine in the presence of different amounts of PANI nanoparticles [51]. The PANI nanoparticles can prevent graphene sheet restacking so as to increase the surface area of the resulting nanocomposites. Compared to rGO only (268 m^2^ g^−1^), the rGO/PANI material showed a high surface area of 891 m^2^ g^−1^. The specific capacitance of the material was found to be 257 F g^−1^ at 0.1 A g^−1^ in 1 M H_2_SO_4_.

Porous micro/nanostructured materials have attracted great interest surrounding their prospects for applications in energy storage. The unique porous structures can enhance supercapacitor performance remarkably, due to high specific surface area and reduced transport lengths for both charge carriers and electrolyte ions [52,53]. Luo et al. prepared hollow-structured graphene-PANI spheres (GSA/PANI HS) via a facile and efficient Pickering emulsion polymerization [53]. The concept is shown in Figure 3g. Amphiphilic sulfonated graphene (GSA) has hydrophobic graphene planes and hydrophilic sulfonic acid, and act as a Pickering stabilizer to stabilize an oil phase. Aniline molecules in oil adsorb to the surface by electrostatic interactions between the amino groups of aniline and the sulfonic groups of GSA, and then polymerize at the surface of the Pickering emulsion droplet. After the oil phase is removed, hollow graphene-polyaniline spheres with a rough outer surface and a smooth inner surface remained (Figure 3a–c). As shown in Figure 3d, aniline polymerized mainly at the outer surface of the GSA stabilized emulsion droplets and formed a large number of PANI particles and fibers. The atomic percentage of sulfur for the outer wall region is lower than that of the inner, which confirms that the PANI formed on the outer wall (Figure 3e,f). This material exhibited a specific capacitance of 546 F g^−1^ at 1 A g^−1^ in 1 M H_2_SO_4_ and enhanced capacitance retention at higher scan rates (76% from 5 to 500 mV s^−1^) and current densities (83.5% from 0.5 to 10 A g^−1^). Compared with hard template methods, this method is facile and environmentally friendly, avoiding the use of acid to remove the template, thereby providing a new way to fabricate hollow-structured materials.

Zhang et al. fabricated rGO/PANI composites with a novel three-dimensional structure for high-performance supercapacitors [54]. As shown in Figure 4a, with the help of steamed water regulation technique, PANI fibers embed into the porous structure of rGO (Figure 4b,c). The reason for this porous structure is that PANI fibers disperse homogeneously in the mixed GO/PANI solution during the experimental process and graphene sheets overlap partially via hydrophobic and π–π interactions during the reduction process. In addition, the intercalated porous structure facilitates electrolyte infiltration which makes full use of electrochemical double layer capacitance of graphene. The hybrid film showed a specific capacitance of 1182 F g^−1^ at 1 A g^−1^ in a three-electrode test cell filled with 1 M H_2_SO_4_. In addition, an assembled symmetric device exhibited a capacitance of 808 F g^−1^ at 1 A g^−1^ and a high energy density (28.06 Wh kg^−1^ at a power density of 0.25 kW kg^−1^).

Hong et al. used an rGO framework created from diffusion-driven layer-by-layer assembly (dd-LbL) as a scaffold for in situ polymerization of aniline within the pores of the framework, to form an rGO/PANI composite [55]. A filter paper coated by branched polyethyleneimine (bPEI) solution was immersed in GO supension. The bPEI then diffused and complexed with the GO sheets. As a result, a 3D porous GO foam formed based on strong electrostatic interactions between GO and bPEI. After that, PANI nanowire array was in situ polymerized on the foam. After hydrothermal treatment, rGO templates were obtained. The 3D porous structure of rGO formed via the dd-LbL assembly not only helped the PANI distribute evenly throughout the whole template, but also played a critical role in preventing graphene restacking, which increases the specific surface area and facilitates ion transportation. This material showed a specific capacitance of 438.8 F g^−1^ at 0.5 A g^−1^ in 1 M H_2_SO_4_. Furthermore, after the the capacitance contributed by the rGO template (C_GO_) was calculated and subtracted from the whole capacitance, they found that the capacitance generated by the PANI nanoparticles (in the presence of an rGO supporting template) is up to 763 F g^−1^.

As shown above, many reports demonstrate that graphene/PANI composites achieve strong synergistic specific capacitance, but researchers do not clearly understand the mechanism behind their high performances. Recently, Zhang et al. investigated the evolution of electrochemical and spectral properties of PANI/graphene [56]. They found that, during electrochemical measurements, the PANI decomposed and degradation products of hydroxyl- or amino-terminated oligoanilines (HAOANIs) were formed (Figure 5a). HAOANIs possess a large specific capacitance (>1000 F g^−1^) higher than PANI, due to the phenol/quinone conversion during the redox process (Figure 5b), however their electric conductivity is very poor. When the electrode material is pure PANI, HAOANIs increase the resistance remarkably resulting in no obvious change in capacitance of PANI. In comparison, when PANI is surrounded by an rGO matrix, the rGO material can provide a conductive path for charge carriers. What is more, the authors propose that a PANI/rGO electrode can be activated at 0.8 V before use, to promote the formation of HAOANI. Following this, the working potential should be limited to remain below 0.7 V and, in this way, a long life cycle could also be realized. As a result, HAOANIs in the PANI/rGO composite led to an increase in the specific capacitance and cycling stability. 

### 2.2. Graphene and PPy Composites

PPy is another significant CP with a lot of appealing characteristics such as, excellent electrical conductivity, good mechanical properties, ease of preparation, and low cost. It has been used in supercapacitors, batteries, and in many other fields outside of energy storage [57,58,59,60]. However, PPy suffers from the same stability issues mentioned for PANI. In order to solve the problem, many researchers proposed the preparation of PPy/carbon composites and that found graphene/PPy composites can remarkably improve the electrochemical performance while exhibiting good mechanical properties.

Most graphene/PPy composites are synthesized by chemical or electrochemical polymerization. In a typical chemical polymerization process, graphene is mixed with the pyrrole monomer, and ammonium persulfate (APS) or FeCl_3_ is used as the oxidant [61,62]. By comparison, electrochemical methods attract many researchers’ interests for their environmental friendliness, without the need for an oxidant. Graphene is a good substrate for the polymerization of PPy as it provides many active sites to adsorb the pyrrole monomer that may then serve as initiation points for polymer formation. 

Recently, Shu et al. fabricated flexible graphene/polypyrrole nanofiber films [63]. Surfactants are adsorbed on the graphene sheets that are difficult to remove, but in this work, the negative effects were minimized. Cetytrimethylammonium bromide (CTAB) was used to exfoliate expanded graphite, and then as a template for growth of nanostructured PPy nanofibers (PPyNF) (Figure 6a,b). When increased to a high scan rate, the CV curves of PPyNF show a leaf shape (Figure 6c), due to the limited ionic and electronic transport. By comparison, neat graphene electrodes maintained their quasi-rectangular CV shape up to 400 mV s^−1^ (Figure 6d). With the incorporation of graphene, the capacitive performance of the as formed G-PPyNF composite film was improved. This is most clearly evidenced by the increased area of the CV loop at high scan rates (Figure 6f). G-PpyNF1 with a high graphene ratio showed a reasonably rectangular CV shape even at 400 mV s^−1^ (Figure 6e). This material showed a specific capacitance of 161 F g^−1^ at 0.5 A g^−1^ in 1 M Li_2_SO_4_ and 80% retention after 5000 cycles. In addition, the film offered a 92% retention of the initial capacitance after 5000 bending cycles. The employed methods not only increased the electrochemical performance, but also enabled flexibility. 

Agglomeration and restacking of graphene sheets are another big problem for researchers producing graphene via top-down methods. To avoid this shortcoming, sulfonated graphene and N-doped graphene have been used to synthesize composites [64,65,66]. Compared with pure graphene, the sulfonated graphene (SG) has a superior electrical conductivity. A highly conducting PPy/SG composite, prepared by interfacial polymerization, achieved a specific capacitance of 360 F g^−1^ at 1 A g^−1^ in 1 M H_2_SO_4_ [65]. A study by Lai et al. showed that N doping of the graphene network could not only improve electrical properties but also facilitate the uniform growth of PPy on both sides of the graphene [64].

Controlling the morphology and structure of composites is the key challenge for enhancing the electrochemical properties. Zhu et al. transformed the typical “cauliflower” morphology of PPy into a homogeneous nano-sheet morphology composite by adjusting the graphene content [67]. This structure provides a more accessible surface area for adsorption of electrolyte ions and reduces ion/charge transport paths. The material exhibited a high capacitance of 255.7 F g^−1^ at 0.2 A g^−1^, and 199.6 F g^−1^ at 25.6 A g^−1^, in 3 M KCl. In addition, it maintained 93% capacitance retention after 1000 cycles at different current densities. 

Zhu et al. synthesized various PPy/rGO composites with different mass ratios by chemical polymerization method using rGO as a support framework [68]. The morphology, specific surface area, and electronic conductivity were optimized by adjusting the content of rGO. Pure PPy agglomerates into a cauliflower morphology, composed of spherical primary particles with a diameter of about 200 nm (Figure 7a,e). After addition of rGO, the PPy tended to form and polymerize on the surface of rGO. As the ratio of rGO to PPy was increased, the morphology progressively changed into a more obviously sheet-like structure (Figure 7b–d,f–h). The SSA of PPy/RGO-10 composites was measured as 126 m^2^ g^−1^ which is 5 times of that for PPy. In addition, this material showed a specific capacitance of 290.1 F g^−1^ at 0.2 A g^−1^ in 3 M KCl. After 20,000 cycles at a current density of 2 A g^−1^, higher than 97.5% capacitance retention was observed.

Core/shell type structures have also been used to improve the properties of composites [69,70]. Qi et al. fabricated a core/shell tubular structured, graphene nanoflake-coated, PPy nanotube (GNF/PNT) hybrid for all-solid-state flexible supercapacitors [70]. The GNF coating not only protected the surface efficiently but also served as the electron transfer pathway; by controlling the amount of GNF coating, the capacitance of the whole composite was able to be optimized. A flexible all-solid-state symmetric supercapacitor device based on this material showed an areal capacitance of 128 mF cm^−2^ at 1.8 mA cm^−2^, with an associated energy density of 11.4 μWh cm^−2^ at a power density of 720 μWh cm^−2^. Cyclic stability was reasonable, with over 80% capacitance retention after 5000 cycles.

Hydrogels and aerogels are created when precursors are assembled into 3D porous materials. These gels have attracted researchers’ attention, because of their large surface areas and good mechanical properties. He et al. fabricated holey graphene/PPy hybrid aerogels (HGPAs) with 3D hierarchical structure by freeze-drying graphene/PPy [71]. The materials showed an interconnected and stable 3D porous network, and PPy nanoparticles uniformly embedded in the aerogel preventing the restacking of holey graphene nanosheets. A HGPA hybrid aerogel electrode with a mass ratio of PPy:HGO = 0.75 exhibited high specific capacitance of 418 F g^−1^ at a current density of 0.5 A g^−1^, outstanding rate capability (80%) at various current densities from 0.5 to 20 A g^−1^ and good cycling performance (74%) after 2000 cycles in 1.0 M KOH aqueous electrolyte. The hierarchical porous structure and synergistic effect between PPy nanoparticles and HG nanosheets had a strong positive influence on the electrochemical performance.

Wu et al. prepared a graphene/PPy hydrogel (PGH) for use in flexible solid-state supercapacitors, via a simple heating approach [72]. The pore structure of PGH not only introduces more electrochemically active surfaces for the absorption/desorption of electrolyte ions, but also provides additional mechanical flexibility. This material exhibited a specific capacitance of 363 F cm^−3^ at a current density of 1.0 mA cm^−3^ and a capacitance retention of 98.6% after 12,000 cycles.

### 2.3. Graphene and Poly(3,4-ethylenedioxythiophene) (PEDOT) Composites

PEDOT is synthesized by polymerization of the 3,4-Ethylenedioxythiophene (EDOT) monomer. With the advantages of good electrical conductivity, large pseudocapacitance, lower toxicity, and easy processability, PEDOT has attracted many researchers’ interests as an electrode material. PEDOT and PEDOT:PSS water solution are two major types [73]. PEDOT is found to be insoluble, and the addition of water-soluble poly(styrene sulfonic acid) (PSS) as the charge-balancing dopant can solve this problem [74].

PEDOT can be synthesized via chemical in situ polymerization, in which sodium polystyrene sulfonate (PSS) is dissolved in HCl or distilled water first, and then EDOT monomer is added under stirring [75,76]. Sometimes, degassing is performed to prevent oxidation due to dissolved oxygen in the water [77]. Next, ammonium peroxydisulfate [(NH_4_)_2_S_2_O_8_)] and iron (III) chloride (FeCl_3_), or sodium persulfate (Na_2_S_2_O_8_) and iron (III) sulfate [Fe_2_(SO_4_)_3_] are added as oxidants to catalyze the polymerization [75,76,77]. In comparison, electrochemical polymerization of PEDOT requires conducting substrates, such as ITO, metals, or carbon material [78,79,80]. 

Neat PEDOT has a specific capacitance from 70 to 130 F g^−1^ depending on the polymerization method employed [81,82]. Moreover, differing morphology, processing parameters, and electrolytic solutions influence the electrochemical performance of PEDOT for supercapacitor applications [73,83]. Compared to other conducting polymers, PEDOT exhibits good cyclic stability in the oxidized state [84], however low specific capacitance limits its application for supercapacitors. Alvi et al prepared G-PEDOT nanocomposites which have a specific capacitance of 304 F g^−1^ in 2 M HCl and 261 F g^−1^ in 2 M H_2_SO_4_ at 10 mV s^−1^ [75]. Another study by Österholm revealed that the redox current and charge values of PEDOT-GO were lower than that of PEDO-PSS, and PEDOT-GO had a smoother surface morphology with less surface area. Aggregation of graphene reduced the transmission efficiency of ions and electrons [85].

Zhou et al. synthesized PEDOT/SDS-GO composites by a facile electrochemical polymerization method where sodium dodecyl sulfate (SDS) served not only as a supporting electrolyte, but also provided counter-ions and acted as a dispersant [86]. The surface of GO sheet is negatively charged, which brings out the driving force for electrodeposition. The PEDOT/SDS had a leaf-like morphology (Figure 8a) and irregular block aggregates, as seen in the TEM image in Figure 8d, but this structure may be unfavorable for the access of electrolyte. On the other hand, PEDOT/SDS-GO films show a petal-shaped morphology (Figure 8b), and this petal-shaped and porous composite film provided a large surface area, limited aggregation, and enhanced ion and electron transport, during charge and discharge (Figure 8g,h). In addition, the sheet-like GO showed a smooth and slightly curly edge which contributed to the large surface area (Figure 8c). PEDOT (red arrows) and PEDOT/SDS particles (blue arrows) formed on the GO surfaces (Figure 8e,f). As a result, PEDOT/SDS-GO showed both double layer capacitance from the GO and pseudocapacitance from the PEDOT/SDS. The films achieved a high areal capacitance of 79.6 mF cm^−2^ at 10 mV s^−1^ and a maximum energy density of 12.5 μWh cm^−2^. 

The assembly of hydrogels is one of the most promising and effective approaches for producing graphene/PEDOT composites with desirable electrochemical properties [87,88,89]. However, weak mechanical properties hinder its application. Zhou et al. prepared a graphene/PEDOT hydrogel with excellent mechanical performance and high conductivity by in situ polymerization [88]. They proved that a high content of carbonyl groups in GO results in a large amount of PEDOT clusters being generated. Due to this homogeneous and compact microstructure, the material showed a good electrical conductivity of 0.73 S cm^−1^ and a specific capacitance of 174.4 F g^−1^. Furthermore, it exhibited an excellent compressive fracture stress of 29.6 MPa and a modulus of about 2.1 MPa at 10 rad s^−1^.

Recently many metal oxides have been used with graphene/PEDOT composites to fabricate ternary composites, as supercapacitor electrodes [90,91,92,93]. The addition of metal oxides can remarkably increase the specific capacitance, while the graphene improves the electrical conductivity. EDOT played the role of reducing agent and monomer, reacting with the highly oxidizing graphene oxide. Moreover, graphene and PEDOT can form a 3D network structure which is favorable for ion transport and increasing the specific surface area. 

### 2.4. Comparison and Summary 

Zhang et al. compared the properties of three different CP-rGO composites. PEDOT, PANI and PPy were singly coated on the surface of rGO sheets via an in situ polymerization to synthesize nanocomposites [94]. A porous surface was observed on the rGO-PANI sample (Figure 9c,d), whereas the surface of rGO-PEDOT was coated by many nanowhiskers (Figure 9a,b). Meanwhile, the rGO-PPy exhibited a flat surface (Figure 9e,f). The CV of both rGO-PEDOT (Figure 9g) and rGO-PPy (Figure 9i) exhibited a rectangular shape. Due to the high conductivity of rGO, the rGO-PEDOT exhibited good capacitive behavior. However, the rGO-PANI sample showed two peaks on the CV curve (Figure 9h). The first peak is associated with the redox transition of PANI from the semiconducting-state to the conductive state, and appears in the range of 0–0.3 V [94,95,96]. The second peak, appearing in the range of 0.4–0.5 V, is due to processes associated with the over oxidation of the polymer, followed by hydrolysis to quinone-type species [94,97]. From the curves it is clear that the voltage window of PANI is from −0.4 to 0.6 V, the voltage window of PPy is from −0.6 to 0.4 V and the voltage window of PEDOT is from −0.1 to 0.8 V. The rGO-PEDOT showed a specific capacitance of 108 F g^−1^ at 0.3 A g^−1^, which was poorer than rGO-PANI (361 F g^−1^) and rGO-PPy (241 F g^−1^) at 0.3 A g^−1^. In contrast, the rGO-PEDOT sample retained 88% of its initial capacitance after 1000 cycles, which was higher than both the rGO-PANI (82%) and rGO-PPy (81%), under the same conditions.

In summary, with its sp^2^ bonding network, graphene provides a substantial number of active sites for the nucleation and growth of CPs, and as a scaffold, it improves the conduction of electrons to and from the CPs. Secondly, 3D gel-type nanostructured composites reduce the volumetric changes of CPs during charging and discharging processes, enhancing the long-term stability. Thirdly, composites with a porous structure increase the accessibility of electrolyte to more surface area, and facilitate fast ion transport. Composite electrodes possessing these qualities are capable of achieving high specific capacitances, which can be attributed to pseudocapacitance contributions from CPs and the electric double layer capacitance of graphene.

## 3. Graphene and Organic Molecule Composites

As discussed above, in the past ten years graphene/CP composites have shown strong performance in research-scale supercapacitors. In recent years, organic molecules which also exhibit pseudocapacitance have been used to modify graphene for electrode materials, and some of these have also shown outstanding performance. With strengths such as low cost, good pseudocapacitance, and flexibility, these organic molecules have been used in batteries, supercapacitors, and many other energy storage fields [98,99].

### 3.1. Graphene and Quinone or Its Derivatives Composites

Among all the potential alternative organic molecules, quinones are well-known as electrochemically active organic molecules, and have been applied as energy storage materials for supercapacitors, Na-ion batteries, and aqueous rechargeable batteries [100,101,102]. However, similar to most of organic molecules, quinones have poor electrical conductivity. In order to solve this problem, quinones are used in combination with highly conductive carbon materials [100,103].

An et al. used anthraquinone (AQ) to functionalize a graphene framework (GF) through non-covalent modification [104]. The synthesized AQ/GF material showed a capacitance of 396 F g^−1^ at 1 A g^−1^, and retained 64% of this capacitance as the current density was increased to a large value of 100 A g^−1^. After 2000 cycles in 1 M H_2_SO_4_, the electrode material still possessed 97% of its initial capacitance. The schematic is shown in Figure 10a. The AQ molecules were adsorbed on the surface of GF through π–π stacking interaction without disrupting the sp^2^ network. This structure led to fast charge transfer reaction and low transfer resistance. Furthermore, due to the parallel aromatic ring of AQ and conjugated carbon skeleton of the GF, the distance between the conductive scaffold and the electrochemical active sites is short. The pure GF is an ideal EDLC material, thus its CV curve was almost rectangular in shape. However, an anodic peak at about −0.087 V and a cathodic peak at about −0.125 V appeared on the CV curve of AQ/GF (Figure 10b). This is because of coupled proton–electron transfer redox reactions (Figure 10c) [104,105]. In addition, the AQ/GF showed both the pseudocapacitance of AQ and EDL capacitance of graphene. The results reveal that non-covalent modification by redox-active organic molecules is a potential way to synthesize supercapacitor electrodes with good properties. 

Boota et al. used 2,5-dimethoxy-1,4-benzoquinone (DMQ) and rGO to synthesize a redox-active aerogel via a hydrothermal method [106]. After fabrication of a binder-free 50 µm thick electrode, this material achieved an excellent specific capacitance of 650 F g^−1^ at 5 mV s^−1^ (780 F cm^−2^) in 1 M H_2_SO_4_ and a remarkable capacitance retention of 99% at 50 mV s^−1^ after 25,000 cycles. There are five reasons for this brilliant performance: (1). The DMQ acted as a spacer, which both minimizes the rGO sheet aggregation and results in a morphology that decreases the macroscopic volume change during cycling. Moreover, the 3D interconnected porous structure provides accessibility to electrolyte ions for fast charge transfer. (2) DMQ offers a high pusedocapacitance by reversible Faradaic processes. (3) Strong π–π interactions between DMQ and rGO enhance the electronic percolation and remain favorable for capacitance retention during the cycle [106,107,108]. (4) Peripheral methoxy groups present at the 2,5-positions of DMQ protect the carbonyl radical intermediate of DMQ, and therefore DMQ degradation upon cycling decreases [106,109]. (5) rGO sheets enhance the mechanical stability and conductivity. These reasons are also at least partially applicable to other organic molecule/graphene composites. The results of DFT calculations support these explanations, suggesting that strong adhesion to the graphene surface with a greater charge redistribution resulted in a longer life cycle, and that the pseudocapacitance of composites mainly originates from the carbonyl groups [106]. These conclusions have great significance for current research. 

### 3.2. Graphene and Other Aromatic Molecules Composites

Molecules containing amino groups are also used to functionalize graphene for energy storage. Lu et al. synthesized the p-phenylenediamine (PPD) modified GO/rGO composites via a covalent approach [110]. The two amine groups of PPD react with functional groups protruding out of the plane of the GO layer, thus graphene sheet separation is able to be maintained and aggregation is prevented [110,111]. There are two routes: “reaction first then reduction” (GPPDH) and “reduction first then reaction” (GHPDD). The specific capacitance of GPDDH is 316.54 F g^−1^ at 10 mV s^−1^, which is larger than that of GHPDD (249.24 F g^−1^). Both of them showed reasonable capacitance retention of 93.66% (GPPDH) and 87.14% (GHPDD) at 2 A/g after 4000 cycles. The result suggested that the GPPDH material contained more PDD inserted into the graphene layers. This prevented the restacking and aggregation of rGO, offering abundant surface for fast reversible faradaic reactions by amino groups of the PPD for pseudocapacitance. In addition, PDD is also known to act as precursor of nitrogen doping [110,112]. By comparison, in the GHPDD film, more PDD molecules were found grafted onto the surface of restacked graphene layers. This paper provided a new idea to combine molecules and graphene, and confirmed amino groups of PPD could improve the electrochemical performance.

The rational selection of organic molecules to functionalize graphene is a problem that troubles researchers. Our group compared the energy storage performance of functionalizing graphene with four different aromatic molecules containing either amino or hydroxyl groups, or both (A: 4,4’-oxydiphenol, B: 4,4’-oxydianiline, C: 3,3’-dihydroxydiphenylamine; and D: diaminobenzidine) (Figure 11a) [113]. Figure 11b shows the redox processes of amino and hydroxyl groups in H_2_SO_4_. Finally, some clues were found: (1) the adsorption onto N-doped graphene (NG) sheets of aromatic molecules with –OH groups was stronger than that of molecules with –NH_2_ groups; (2) more redox-active groups on a single molecule does not necessarily mean it provides higher pseudocapacitance; (3) –NH_2_ groups have better cyclic stability and ability to sustain a wider voltage window than that of –OH groups, in Li_2_SO_4_ electrolyte. In addition, we found 4,4’-oxydiphenol functionalized NG films achieved an excellent specific capacitance of 612 F g^−1^ at 5 mV s^−1^, in a symmetric cell with Li_2_SO_4_ electrolyte. A summary of information about recent works reporting on graphene modified with organic molecules for supercapacitors, is shown in Table 1.

As mentioned above, most organic molecules can be used to functionalize graphene by non-covalent or covalent methods. Each has its own advantages; the covalent approaches may be easy to control and are stable, but in non-covalent methods organic molecules adsorb on the surface of graphene by π–π stacking interactions [121]. When compared, the latter approach results in a structure with more sp^2^ hybridization of carbon atoms in the graphene, so it may possess a better conductivity [104]. Furthermore, the ratio of organic molecules to graphene, the different groups of molecules, and many other factors, have a great influence over the electrochemical performance of the materials when applied to supercapacitors. Although great results have been achieved with the use of organic molecules for energy storage, only a small number of potential candidate molecules ave been studied and with limited understanding of the mechanisms for charge storage. Therefore, further exploration of organic molecules used in this context presents considerable future challenges and opportunity for researchers.

## 4. CPs and Graphene Composites for Novel Devices

### 4.1. Flexible Supercapacitors

In recent years many flexible energy storage devices have been investigated and designed by researchers for applications in flexible displays, smart electronics, medical devices and other flexible devices. Given the advantage of being highly integratable and potentially even wearable, flexible energy storage architechture is considered to be one of the most prominent emerging disruptive-technologies that may completely transform our lives [122,123,124]. Many carbon materials such as CNTs, carbon fibers and graphene are used to fabricate flexible supercapacitor electrodes [125,126,127,128,129,130]. Among them, CPs and graphene composites show outstanding behaviors.

Typical flexible CP/graphene supercapacitors not only have excellent electrochemical properties, but also exhibit favorable mechanical performance. There are five commonly explored methods for fabricating flexible composite electrodes; electrodeposition, in situ polymerization, direct coating, chemical vapor deposition (CVD), and the vacuum filtration technique [131]. 

Shu et al. synthesized rGO-PPy hybrid paper for flexible supercapacitors by electropolymerization on a paper-like graphene gel [132]. This hybrid paper showed a high areal capacitance of 440 mF cm^−2^ at 0.5 A g^−1^ in 1 M H_2_SO_4_. Because of the high electrical conductivity and good chemical stability, graphene papers have been used as a flexible substrate for electrodes in recent years, but it is difficult to fabricate large-scale graphene paper materials via vacuum filtration. Inkjet printing technique address this challenge. Chi et al. used well-controlled full inkjet printing method to fabricate a freestanding graphene paper (GP) supported 3D porous graphene hydrogel-polyaniline (GH-PANI) nanocomposite [133]. GO ink was printed on a commerical paper, followed by overprinting the GH-PANI inks on it (Figure 12a). The graphene hydrogel showed a 3D porous network driven by π–π stacking interaction (Figure 12b), with coral-like PANI loaded onto the pore walls of the graphene (Figure 12c). The final layered stack of GP and porous GH-PANI is shown in Figure 12d. Compared to other methods, this low-cost and efficient technique not only produced large-area flexible and lightweight graphene-PANI nanohybrid paper, but also took advantage of the synergistic effect of PANI and graphene. The flexible all-solid-state symmetric supercapacitor based GH-PANI/GP exhibited an energy density of 24.02 Wh kg^−1^ at 400.33 W kg^−1^ combined with remarkable mechanical flexibility.

In the traditional fabrication of supercapacitor electrodes, active materials are mixed with carbon black and a binder (such as PVDF or PTFE). However, binder materials are generally non-conducting and therefore decrease the electrical conductivity of electrodes, inhibiting electrochemical performance. For flexible supercapacitors the graphene may act as a flexible provider of mechanical strength, and as an active charge-storing part of composites. Furthermore, PANI or PPy, for example, are able to be polymerized directly on the graphene surface, without any binder, which is a good way to reduce the mass of material in an electrode which is not contributing to the electrochemical performance.

The bending test is a common method to judge the flexibility of supercapacitors. The intention is to ensure that no cracks or fractures appear after repeated bending at different angles during cycling, as these will detrimentally affect the electrochemical performance. However, materials satisfying only the requirements of the bending test may still not be robust enough to satisfy the needs of flexible applications. Twisting and folding properties, without loss of performance, may also be deemed necessary for supercapacitors being designed for flexible applications. Li et al. prepared a flexible and free-standing graphene/polyaniline (G/PANI) composite electrode [134]. The G/PANI composite nanosheets were embedded in the skeleton of the 3D graphene framework and wrapped by graphene sheets (Figure 13a), which not only reduced the stacking of graphene but also provided favourable mechanical performance (Figure 13b,c). This material exhibited a high specific capacitance of 777 F g^−1^ and 990 F cm^−3^ at 1 A g^−1^ in 1 M H_2_SO_4_ and 85% capacitance retention after 60,000 cycles in a three-electrode cell configuration. Out of this 3D-G/PANI composite film, flexible all-solid-state supercapacitors (ASS) were prepared, without the need for any current collectors or supporting substrates (Figure 13d). The ASS showed remarkable mechanical flexibility and durability (Figure 13e). Moreover, under bending, twisting, and even folding, the electrochemical performance of ASS didn’t change (Figure 13f). In the folded state, the ASS showed a remarkable specific capacitance of 665 F g^−1^ and 847 F cm^−3^, with an excellent capacitance retention of 86% at 20 A g^−1^ after 10,000 cycles.

Xie et al. synthesized graphene/PANI composite film with a wavy shape, able to behave as a stretchable upercapacitance electrode [135]. Porous graphene was grown on a Ni foam template which was later removed by acid etching, and then following this, a thin film of PANI was deposited on the surface of the graphene by pulsed electrochemical deposition. The porous graphene had some wrinkles and even cracks, however the thin PANI film coated on the surface of it improved the mechanical strength. Due to its structure, the materials can be stretched up to 30% and bent from −180° to 180°. When the electrode material was fabricated into an all-solid-state supercapacitor, the device showed a specific capacitance of 261 F g^−1^ and an energy density of 23.2 Wh kg^−1^ at a power density of 399 W kg^−1^. Furthermore, in the bent or stretched state, the specific capacitance and capacitance retention are close to the normal state, demonstrating that this wavy shape device can tolerate mechanical deformation and maintain good electrochemical performance at the same time.

Recently, Ren et al. combined CVD and chemical interfacial polymerization methods, to fabricate stretchable electrodes based on a graphene foam/PPy [136]. Pure graphene foam showed a 3D interconnected network structure with a smooth surface (Figure 14a,b) and when PPy nanoparticles were added, these grew uniformly on the surface of foam (Figure 14c,d). The stretchable supercapacitor based on this material could be stretched to 10%, 20%, 30% and 50% strain without substantial changes in CV curves and GCD curves (Figure 14e–g). When stretched to these amounts the supercapacitor showed areal specific capacitance retentions of 98%, 92%, 84%, and 69%, respectively. After 100 charge/discharge cycles the capacitance retention of the supercapacitor under 30% strain, was 88% (Figure 14h).

Compared with the liquid electrolytes commonly used in supercapacitor research, flexible devices often use a PVA based gel impregnated with electrolytes. This provides the possibility of omitting the rigid packaging that energy storage devices generally require to prevent electrolyte leakage. Therefore, the development of flexible leak-proof electrolyte systems is also a critical aspect of producing flexible and/or wearable energy storage, however detailed discussion of such systems is beyond the scope of this review which is centred on electrode materials. Nevertheless, when these components are brought together the uniquely novel mechanical properties of flexible and stretchable supercapacitors show strong potential as a disruptive technology to a wide range of established markets.

### 4.2. Wearable Supercapacitors

In the above discussion, flexible supercapacitors are presented as an emerging energy storage device architecture with highly desirable mechanical properties. Wearable energy storage generally refers to devices fabricated within flexible and lightweight fibers or yarns, which are then woven into textile fabrics. To satisfy the needs of wearable energy storage, devices must not only show excellent electrochemical performance and possess all the mentioned flexible behaviors (bending, folding, and stretching) but must also have added durability in order to resist the destructive forces that worn textiles are exposed to in application. 

Huang et al. synthesized stainless steel yarns coated with a PPy@MnO_2_@rGO hierarchical structure via electrodeposition [137]. The process is schematically represented in Figure 15a. The fiber-based supercapacitors fabricated from this modified yarn showed an outstanding specific capacitance of 486 mF cm^−2^ in 1 M Na_2_SO_4_ (three-electrode cell), and 411 mF cm^−2^ in an all-solid-state two-electrode cell with PVA/H_3_PO_4_ electrolyte. Mechanical deformation tests were performed, consisting of bending at 90°, knotting, and twisting, each repeated 1000 times. Following each of these tests, the devices had retained capacitances of 80%, 91% and 103%, respectively. In a practical demonstration of the devices’ capabilities, 7 yarn supercapacitors were woven into a 15 × 10 cm^2^ piece of cloth, connected in series, and then used to light 30 LEDs. This provides clear evidence that yarn-based supercapacitors have the potential to possess excellent electrochemical and mechanical properties, giving them (literally) a bright future in various wearable electronics.

Although the electrochemical performance of yarn supercapacitors has been established, it still lags behind the energy requirement for conceivable applications, such as heated clothing, or charging personal devices. In fact the amount of charge stored via the EDL capacitance mechanism is proportional to the surface area of an electrode interfacing with an electrolyte. In the case of yarn supercapacitors, it may not be clear how this can be increased without either longer or thicker yarn. If it were possible to create additional inner interface (i.e. within the core of the yarn), the specific capacitance of yarn supercapacitors could be increased. This is what researchers have set out to do.

Qu et al. fabricated a hollow graphene/CP fiber electrode with a high energy density when tested as a yarn supercapacitor [138]. As shown in Figure 15c, PEDOT: PPS, GO and vitamin C were mixed to synthesize hollow composites [139]. The addition of PEDOT:PPS aids the GO in formimg lyotropic nematic liquid crystals which are beneficial for fiber formation. This fiber-based supercapacitor achieved a specific areal capacitance of 304.5 mF cm^−2^ at 0.08 mA cm^−2^ and a high energy density of 27.1 μWh cm^−2^ at a power density of 66.5 μW cm^−2^. Due to the nematic state of rGO sheets which arrange along the length of a fiber, and π–π interactions between rGO and PEDOT, the fiber showed an outstanding tensile strength of up to 631 MPa. After bending 500 times the CV curve was almost same as the initial curve. It is suggested that the hollow structure not only increases the capacitance but also provides favourable flexibility and stability.

High capacity performance and robust mechanical properties are both essential characteristics for wearable supercapacitor electrodes. In some cases, to improve the mechanical properties, flexible textile substrates were chosen [140,141]. However, this can have complications; power handling may be relatively low due to poor conductivity of the textile, while limited adhesion between the textile substrate and active materials may result in the electrochemical performance dropping away. A “dyeing and drying” strategy was used to fabricate PPy/graphene/SnCl_2_ modified polyester textile electrodes, as shown in Figure 15b [142]. An rGO layer is uniformly painted on the surface of SnCl_2_ modified polyester fiber (M-PEF) and then PPy is deposited on the substrate of graphene/SnCl_2_ modified polyester yarn (rGO/M-PEFY). The large-area textile electrodes and ultralong yarn electrodes fabricated by this material showed excellent electrochemical and mechanical properties. This material showed an outstanding areal capacitance of 1117 mF cm^−2^ at a current density of 1 mA cm^−2^ in 1 M Na_2_SO_4_ (three-electrode cell) and a capacitance retention of 100% after 10,000 cycles at 20 mA cm^−2^. The textile electrode achieved a capacitance retention of 98.3% when bent from 0° to 180° after 1000 cycles. The polypyrrole/graphene/SnCl_2_ modified polyester textile (PPy/rGO/M-PEFT) electrodes exhibited outstanding electrochemical performance, which is attributed to the synergistic effect of the pseudocapacitive PPy, and highly conductive electric double layer forming rGO. The results reveal this method is suitable for fabricating textile and yarn electrodes, with both the desirable electrochemical and mechanical behaviors demanded by wearable electronics.

Graphene/CP hydrogels with 3D structure exhibit a lot of electrochemical advantages, yet poor mechanical properties can limit their utility for flexible applications. This is because embedding CPs into a graphene hydrogel may bring added challenges to the morphology control, and furthermore, poorly adhering CP active materials may cause cracking in hydrogel structures during mechanical deformation, resulting in electrical discontinuity. Recently, Li et al. reported on 3D graphene/nanostructured conductive polymer hydrogels (CPHs) produced via a self-assembly technique, at room temperature [143]. PANI nanoparticles were wrapped into GO sheets, then after reduction, a nanostructured 3D hydrogel is formed consisting of a network of interconnected PANI and 2D rGO nanosheets. The authors state that the hybrid hydrogels may be molded into fibers via this strategy, but it is unclear from their report how this was achieved. As a result, a fiber-shaped hydrogel electrode showed a volumetric energy density of 8.80 mWh cm^−3^ at the power density of 30.77 mW cm^−3^. There were no changes in the CV curves under normal, knotted, and twisted states (Figure 16a). In addition, capacitance retention changed only subtly when changing stretching strains, and after repeated stretching cycles (Figure 16b,c). Figure 16d shows that when several devices are connected in series to elevate the voltage, that no loss in discharge duration is observed. Moreover, a self-supported all-hydrogel-state textile successfully powered two tandem LEDs in the normal and curved state (Figure 16e). The results reveal that this material can accomodate the large structural deformations that can be expected to affect a wearable energy storage device. 

In the above studies, graphene and CPs are successfully used to fabricate yarn electrodes and woven textile electrodes for wearable supercapacitors. Graphene not only acts as a highly conductive, electric double-layer capacitance material, but also connects the CPs with a flexible, highly conductive scaffold. On the one hand, the CPs contained in the reviewed composite electrodes contribute pseudocapacitance to the electrochemical performance. On the other hand, they play an important role in the porous structure formation of graphene and may also work as a coating to protect the inner material. The synergistic effect of graphene and CPs effectively improves the electrochemical and mechanical properties of textile based devices for wearable electronics. With continued development of textile energy storage technology, it is anticipated that supercapacitors will be integrated into commercialized wearable electronics within the next few years.

## 5. Conclusions and Future Perspectives

As low-cost, high pseudocapacitance materials, CPs have attracted lots of attention in next generation supercapacitor electrode research. While pure CPs show many problems, such as relatively poor electrical conductivity, poor cyclic stability, and low energy density, the combination of graphene and CPs demonstrates how these problems may be solved. This paper has reviewed graphene and CPs including PANI, PPy and PEDOT, as composite electrodes for supercapacitors. Table 2 shows a summary of highlighted results for recent graphene and CP composites for supercapacitors, which have been presented within this review. 

It has been shown that graphene/CP composites may be constructed to possess desirable 3D porous structures. In summary of the synergistic effects, graphene is able to reduce the volume change during charge/discharge cycles, improving the cyclic stability of CPs via reducing the mechanical stresses on them, while at the same time the graphene improves the electrical conductivity and resultant power handling. Meanwhile, CPs can be grown in situ on the surface of graphene which typically reduces the aggregation of the GO precursor during reduction processes, maintaining higher interfacial area within the electrode for improved double layer capacitance. Furthermore, under certain synthesis conditions the excellent mechanical properties of some graphene/CPs composite films demonstrate that the materials can be drawn into yarns suitable for flexible and wearable energy storage devices. 

In recent years, the functionalization of graphene with organic molecules has been investigated for energy storage. There are a number of reports where functionalized graphene based electrodes show outstanding specific capacitances due to the high pseudocapacitance contribution of the organic molecules. Although the number of reports discussing organic molecules for energy storage devices is growing, up to now, only a few amino or hydroxyl group-containing molecules have been investigated, leaving many other molecules remaining to be studied. In addition, details of the energy storage mechanisms of these molecules is still to be clearly understood. 

In most cases discussed, the reported high energy densities and favourable specific capacitances were obtained using small amounts of active materials as electrodes in lab-scale test cells and/or devices. However, for practical applications, supercapacitor devices are typically much larger and require more active materials constructed into larger electrodes. Scaling-up the production of many of the active material combinations discussed in this review is a serious challenge for researchers and engineers. In addition to this broad limitation, the composites in most of the cited literature can only be used in aqueous electrolyte systems, whose potential window is limited to around 1.2 V. This may be compared to organic-solvent or ionic liquid electrolyte systems which may reach around 3 V. Although the relationship has not been discussed in detail, lower maximum voltage severely limits the energy density of devices. While aqueous systems may find some genuine niches, such as for biological implants, the relatively lower energy density generally limits their application. Thus, to improve the energy able to be stored in supercapacitors, electrolyte systems that are able to support a wider potential window are always being sought, especially for graphene/CP composites where the present choices are more restricted by the chemistry of the active materials. 

In conclusion, graphene/CP composites are promising materials for supercapacitor applications. However, currently the available CPs for supercapacitors are limited to PANI, PPy, and PEDOT. Hence their drawbacks in terms of poor cycling stability and narrow voltage window in aqueous electrolyte are anticipated to be solved by the synthesis of new-type CPs. Building asymmetric supercapacitors by using two dissimilar CPs, one as the negative electrode and the other one as the positive electrode, can be an efficient way towards high performance supercapacitors with high specific capacitance and broad voltage window beyond the crytical voltage of water splitting (~1.23 V). With continued effort made to overcome the identified obstacles, and continued research into new-type CPs with better cycling stability, higher specific capacitance, and broader voltage window in aqueous electrolyte, completely novel energy storage devices based on graphene/CPs composites could become commercially available in the not-too-distant future.

## Figures and Tables

**Figure 1 materials-12-00703-f001:**
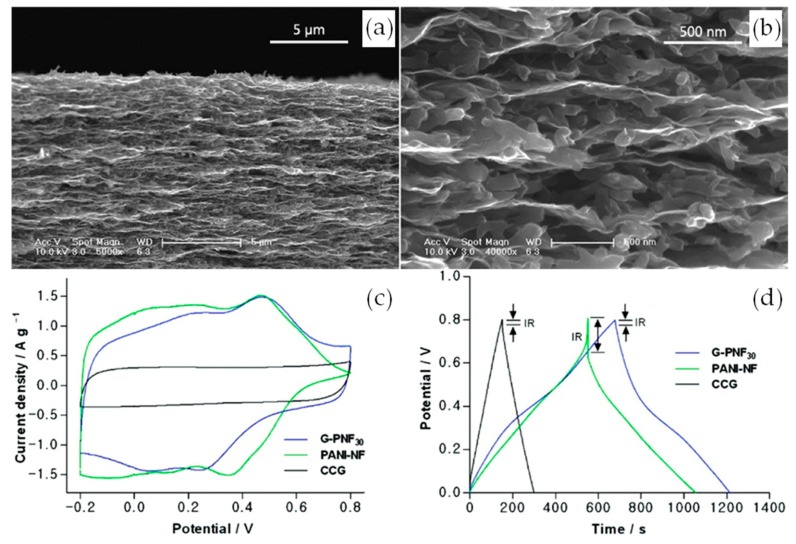
(**a**,**b**) Cross-sectional SEM images of G-PNF_30_; (**c**) CV curves of supercapacitors based on G-PNF_30_, as-formed PANI-NF, and CCG films; (**d**) GCD curves of the supercapacitors based on G-PNF_30_, as-formed PANI-NF, and CCG films. Reproduced with permission from [47]. American Chemical Society (2010).

**Figure 2 materials-12-00703-f002:**
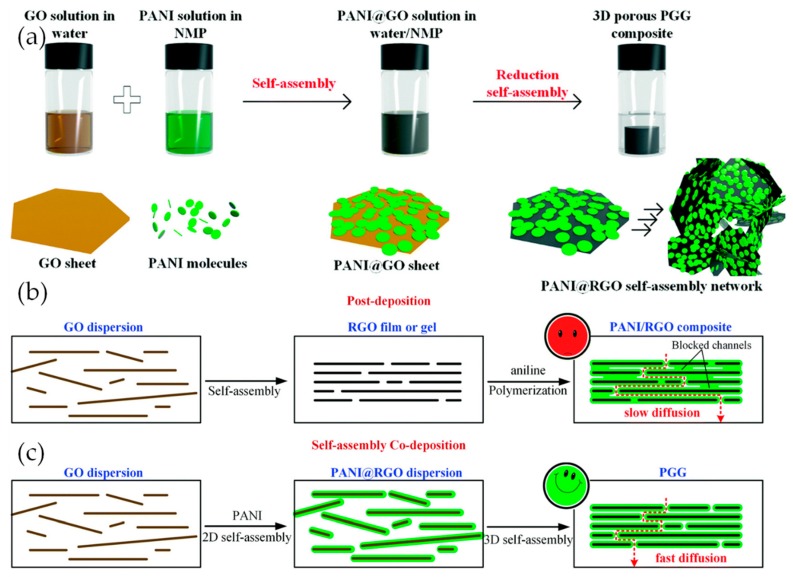
(**a**) Schematic illustration of the solution-based self-assembly method for the preparation of PGGs; (**b**) Schematic illustration of the formation of partially blocked channels in in situ polymerization of aniline on an RGO matrix; (**c**) Schematic illustration of the formation of unblocked channels in a self-assembly process. Reproduced with permission from [48]. The Royal Society of Chemistry (2018).

**Figure 3 materials-12-00703-f003:**
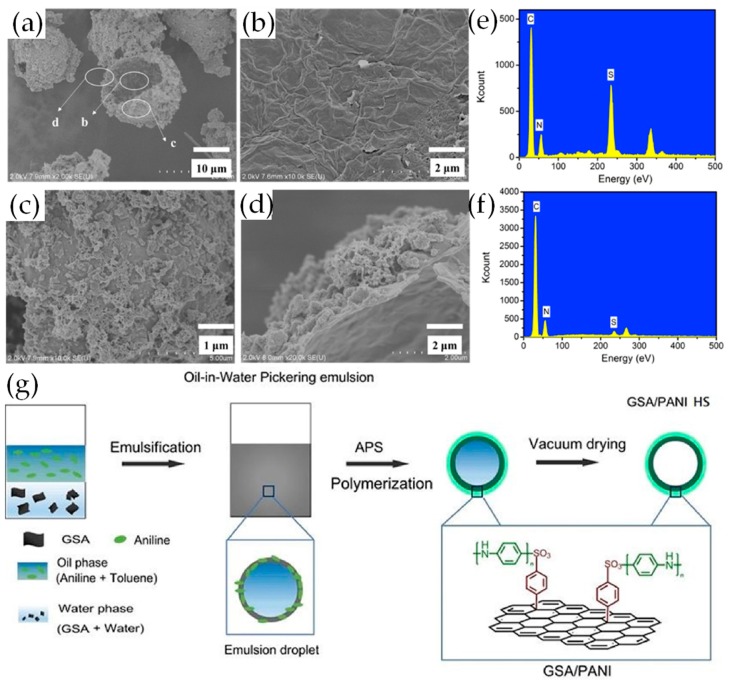
(**a**) GSA/PANI HS FE-SEM images of; FE-SEM images of (**b**) inner wall, (**c**) outer wall and (**d**) edge of GSA/PANI HS; (**e**) EDX spectrum of the regions of GSA/PANI HS inner wall; (**f**) EDX spectrum of the regions of GSA/PANI HS outer wall; (**g**) Schematic illustration of the fabrication of GSA/PANI HS via Pickering emulsion route; Reproduced with permission from [53]. Elsevier (2018).

**Figure 4 materials-12-00703-f004:**
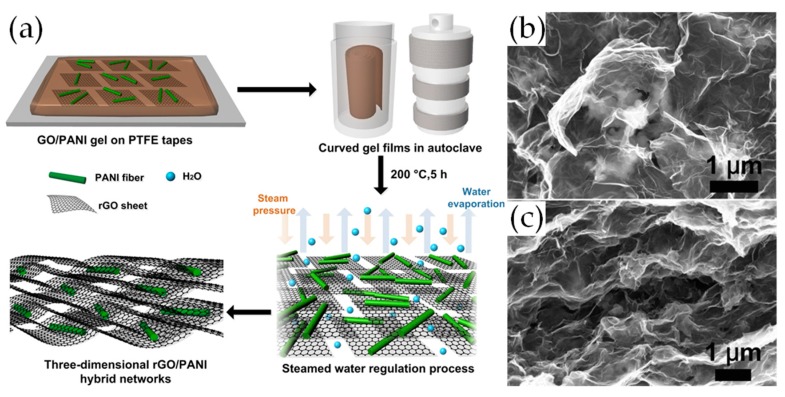
(**a**) Schematic illustration of rGO/PANI hybrid films by steamed water regulation techniques; (**b**) FE-SEM images of rGO/PANI (50%) hybrid films; (**c**) Cross-section FE-SEM images of rGO/PANI (50%) hybrid films. Reproduced with permission from [54]. Elsevier (2017).

**Figure 5 materials-12-00703-f005:**
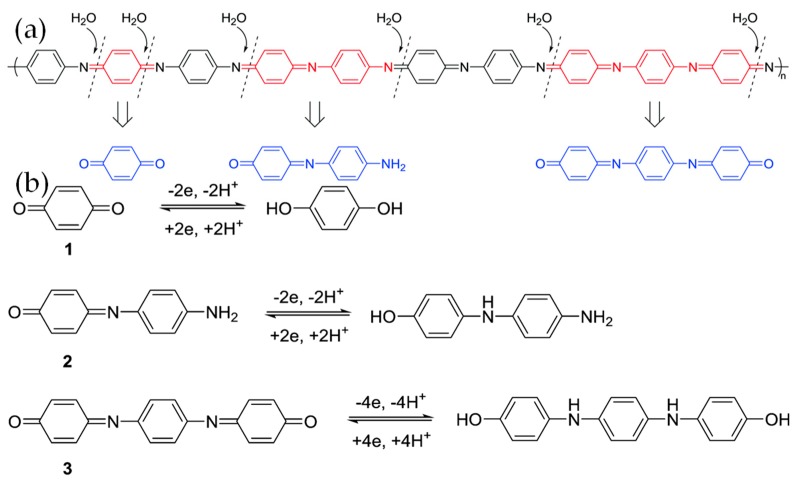
(**a**) Degradation of PANI during the electrochemical process; (**b**) Reversible redox reaction of HQ and HAOANIs. Reproduced with permission from [56]. The Royal Society of Chemistry (2018).

**Figure 6 materials-12-00703-f006:**
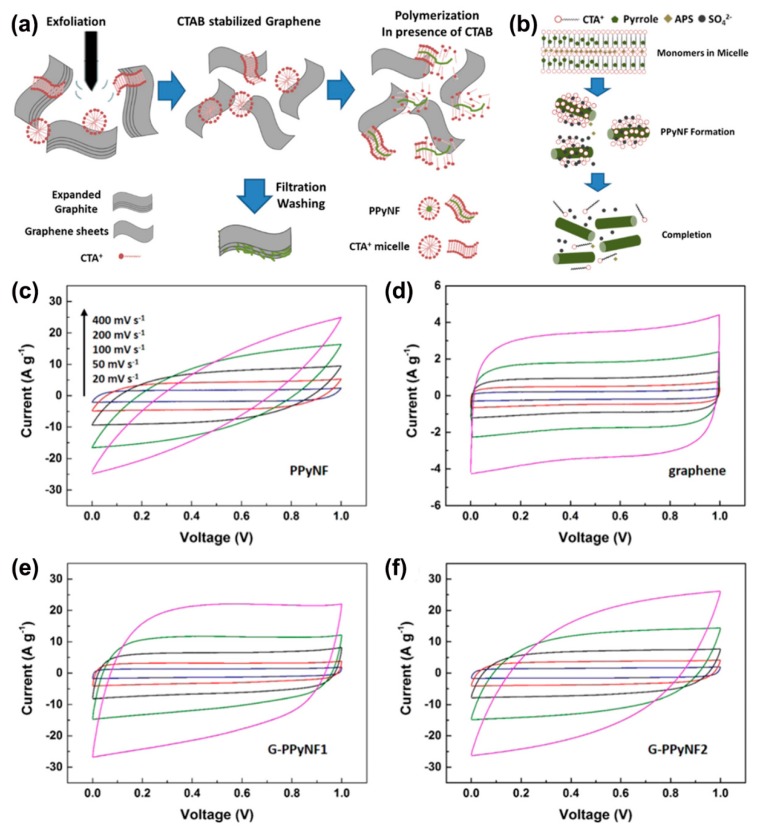
(**a**) The preparation scheme of G-PPyNF film; (**b**) Schematic illustration of the formation of PpyNF; CV curves of PPyNF (**c**), graphene (**d**), G-PPyNF1 (**e**), and G-PPyNF2 (**f**) electrodes. Reproduced with permission from [63]. American Chemical Society (2018).

**Figure 7 materials-12-00703-f007:**
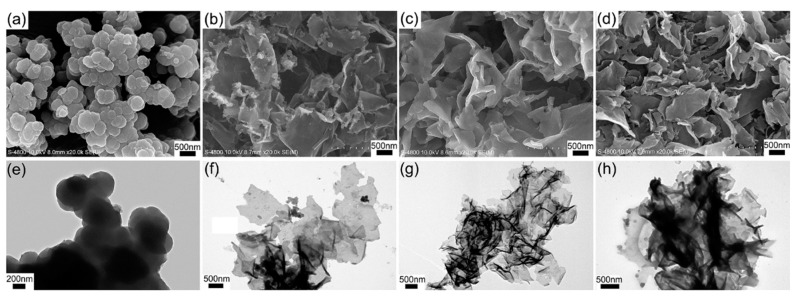
SEM and TEM images of PPy (**a**,**e**); PPy/rGO-5 (**b**,**f**); PPy/RGO-10 (**c**,**g**); and PPy/RGO-15 (**d**,**h**). Reproduced with permission from [68]. Elsevier (2018).

**Figure 8 materials-12-00703-f008:**
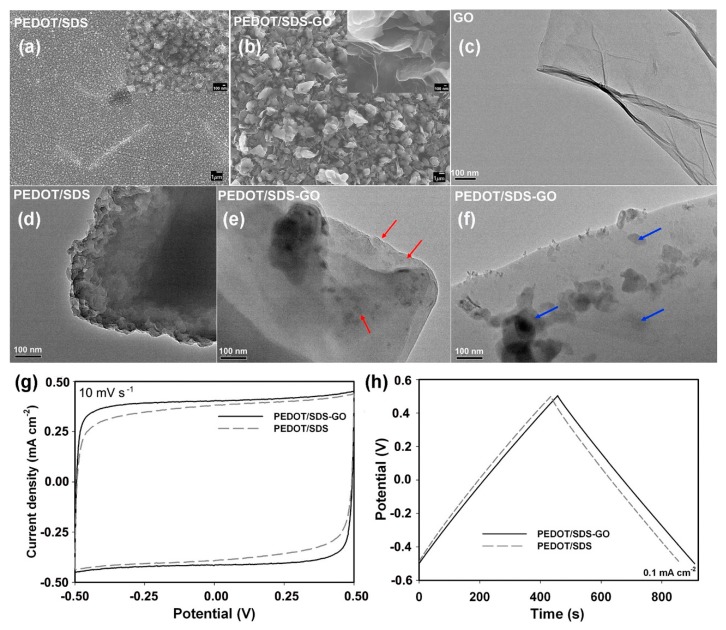
SEM images of PEDOT/SDS (**a**) and PEDOT/SDS-GO (**b**) composite films; TEM images of GO (**c**), PEDOT/SDS (**d**), and PEDOT/SDS-GO composites (**e**), (**f**). Reproduced with permission from [86]. Elsevier (2018).

**Figure 9 materials-12-00703-f009:**
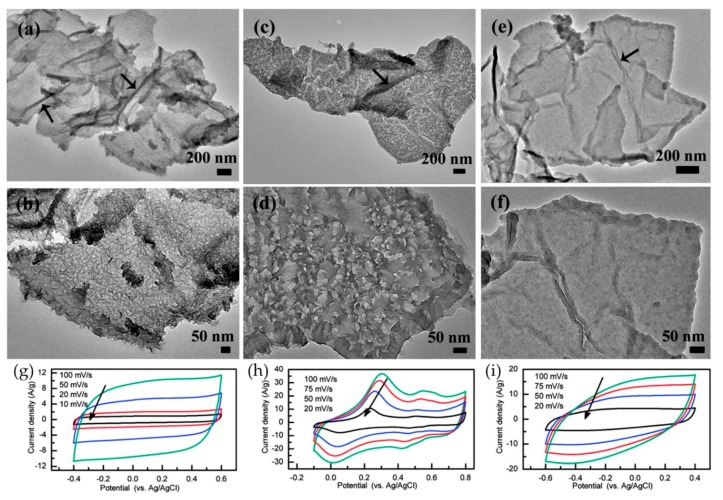
TEM images of rGO-PEDOT (**a**,**b**), rGO-PANi (**c**,**d**), and rGO-PPy (**e**,**f**); CV curves of rGO-PEDOT (**g**), rGO-PANi (**h**), and rGO-PPy (**i**). Reproduced with permission from [94]. American Chemical Society (2012).

**Figure 10 materials-12-00703-f010:**
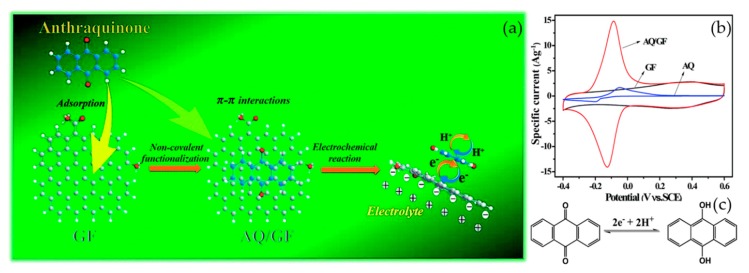
(**a**) Schematic illustration of AQ/GF composite; (**b**) CV curves of AQ, GF and AQ/GF; (**c**) The corresponding redox reactions by a coupled proton–electron transfer. Reproduced with permission from [104]. The Royal Society of Chemistry (2015).

**Figure 11 materials-12-00703-f011:**
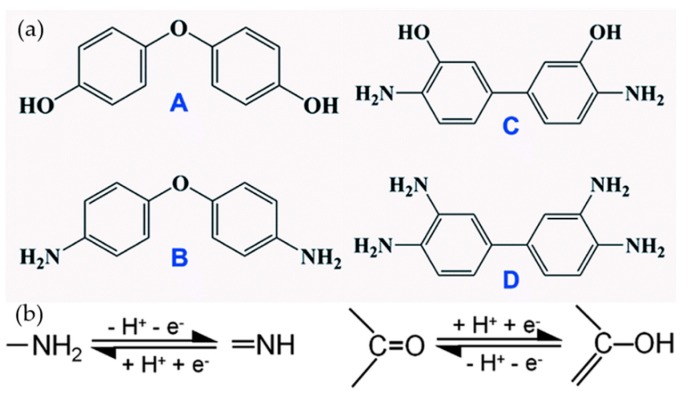
(**a**) Molecular structures of four types of small aromatic molecules; (**b**) The redox processes of amino and hydroxyl groups in H_2_SO_4_ electrolyte. Reproduced with permission from [113]. The Royal Society of Chemistry (2018).

**Figure 12 materials-12-00703-f012:**
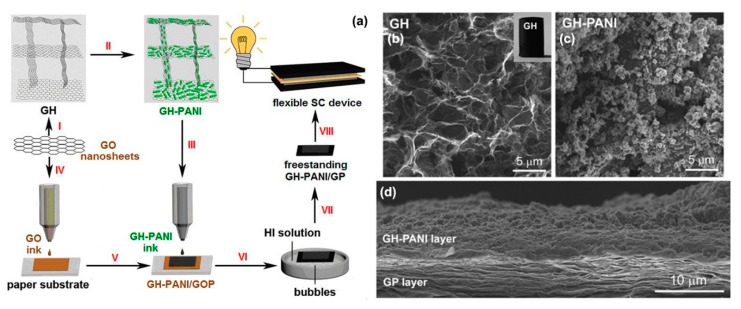
(**a**) Schematic illustration of the fabrication process of GH PANI/GP; SEM images of GH (**b**); GH-PANI (**c**); GH-PANI/GP (**d**). Reproduced with permission from [133]. American Chemical Society (2014).

**Figure 13 materials-12-00703-f013:**
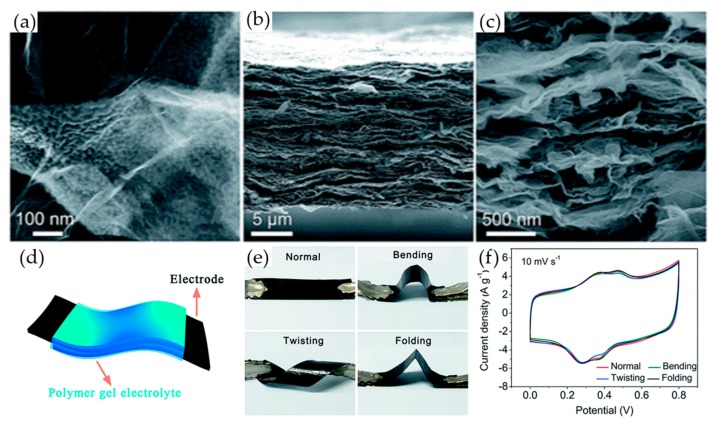
(**a**) SEM images of the 3DG/PANI composite; (**b**) Low- and (**c**) high-magnification SEM side view of the flexible 3D-G/PANI composite film; (**d**) Schematic illustration of a flexible 3D-G/PANI composite-based ASS; (**e**) Digital photographs of the flexible ASS under normal, bent, twisted and folded states; (**f**) CV curves of the flexible ASS under different deformation conditions. Reproduced with permission from [134]. The Royal Society of Chemistry (2017).

**Figure 14 materials-12-00703-f014:**
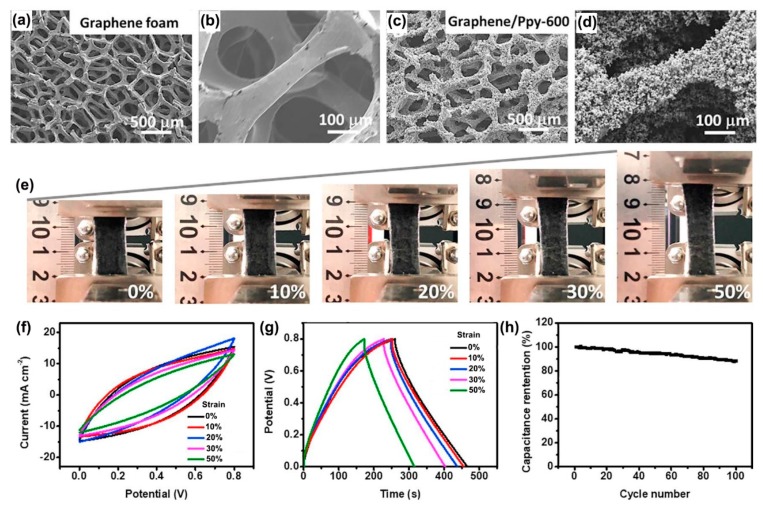
SEM images of (**a**,**b**) graphene foam; (**c**,**d**) graphene/PPy-600 at different magnifications; (**e**) digital photographs of a graphene/PPy-600 all-solid-state supercapacitor being stretched from 0% to 50% strain; (**f**) CV curves of the supercapacitor under different tensile strain; (**g**) GCD curves of the supercapacitor under different tensile strains; (**h**) dependence of the normalized capacitance on the number of stretching cycles for a tensile strain of 30%. Reproduced with permission from [136]. Elsevier (2018).

**Figure 15 materials-12-00703-f015:**
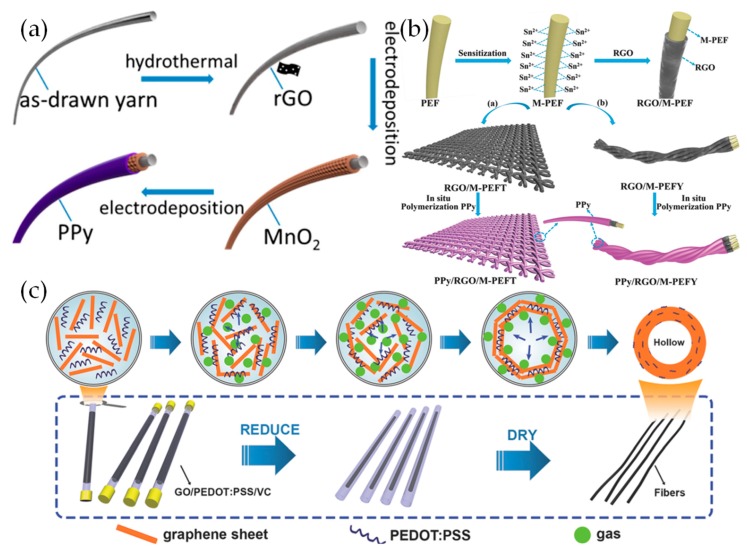
(**a**) Schematic illustration of the yarn modified by deposition of rGO, MnO_2_ and PPy. Reproduced with permission from [137]. American Chemical Society (2015). (**b**) Schematic illustration of the synthesis route toward of textile electrodes and yarn electrodes. Reproduced with permission from [142]. Wiley (2018). (**c**) The preparation scheme for hollow structure composites. Reproduced with permission from [138]. Wiley (2016).

**Figure 16 materials-12-00703-f016:**
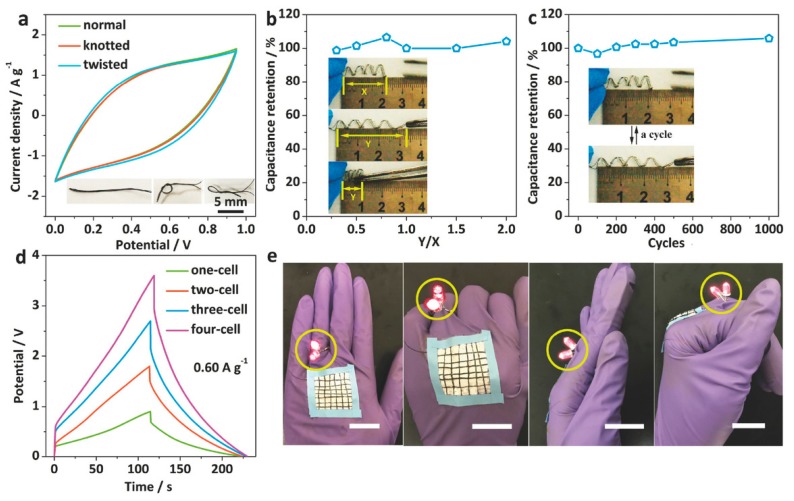
(**a**) CV curves of the assembled all-hydrogel-state cell in different states; (**b**,**c**) the elasticity of the spring-like full cell; (**d**) several full cells connected in series; (**e**) a demonstration of yarn powering two LEDs in the normal and curved states. Reproduced with permission from [143]. Wiley (2018).

**Table 1 materials-12-00703-t001:** Performance of organic molecule/graphene composite supercapacitors.

Year	Materials	Electrolyte	Capacitance	Retention (Number of Cycles)	References
2017	RGO-AQDS	1 M H_2_SO_4_	298.5 F g^−1^ at 1 A g^−1^	82% (10,000)	[100]
2015	AQ/GF	1 M H_2_SO_4_	396 F g^−1^ at 1 A g^−1^	97% (2000)	[104]
2016	DMQ@rGO	1 M H_2_SO_4_	650 F g^−1^ at 5 mV s^−1^	99% (25,000 at 50 mV/s)	[106]
2017	DT-RNGs	1M H_2_SO_4_	491 F g^−1^ at 1 A g^−1^	98.8% (10,000)	[114]
2017	THAQ/rGO	1 M H_2_SO_4_	259 F g^−1^ at 1 A g^−1^	97.9% (10,000 at 20 A/g)	[115]
2015	GPPDH	1 M H_2_SO_4_	316.54 F g^−1^ at 10 mV s^−1^	93.66% (4000 at 2 A/g)	[110]
2018	Graphene/4,4’-oxydianiline	1 M H_2_SO_4_	612 F g^−1^ at 5 mV s^−1^	95% (5000)	[113]
2015	AZ-SGHs	1 M H_2_SO_4_	350 F g^−1^ at 1 A g^−1^	88% (1000)	[116]
2015	Ap-rGO	6 M KOH	160 F g^−1^ at 5 mV s^−1^	85% (5000)	[117]
2018	PTY-NH2/rGO	1 M H_2_SO_4_	326.6 F g^−1^ at 0.5 A g^−1^	90% (4000)	[118]
2018	AQS@rGO	1 M H_2_SO_4_	567.1 F g^−1^ at 1 A g^−1^	89.1% (10,000)	[119]
2014	BPA/RGO	1 M H_2_SO_4_	466 F g^−1^ at 1 A g^−1^	90% (4000)	[120]

**Table 2 materials-12-00703-t002:** Performance of CPs/graphene composite supercapacitors.

Year	Materials	Electrolyte	Capacitance	Retention (Number of Cycles)	References
2018	PANI/graphene	1 M H_2_SO_4_	808 F g^−1^ at 53.33 A g^−1^	-	[48]
2018	GSA/PANI HS	1 M H_2_SO_4_	546 F g^−1^ at 1 A g^−1^	-	[53]
2017	rGO/PANI	1 M H_2_SO_4_	1182 F g^−1^ at 1 A g^−1^	~100% (10,000)	[54]
2018	G-PPyNF	1 M Li_2_SO_4_	161 F g^−1^ at 0.5 A g^−1^	80% (5000)	[63]
2018	PPy/rGO	3 M KCl	290 F g^−1^ at 0.2 A g^−1^	97.5% (20,000)	[68]

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
