# Peer review of "A Review of Supercapacitors Based on Graphene and Redox-Active Organic Materials"

_materials, 2019, doi:10.3390/ma12050703_

Round 1
Reviewer 1 Report
This review manuscript covers wide range of electrode materials consisting of graphene, organic molecules, and conducting polymers. I think this manuscript was well written, and introduced important strategies, and concepts for graphene-based hybrid electrodes. I suggest this manuscript to be published without any revision. Just a minor correction is needed in page 3, line 86, CP should be used instead of conducting polymer.
Author Response
Point 1: I suggest this manuscript to be published without any revision. Just a minor correction is needed in page 3, line 86, CP should be used instead of conducting polymer.
Response: We have corrected this problem.
Reviewer 2 Report
This review paper describes an overview of graphene and redox-active organic materials for application in supercapacitors. It seems that the authors did not invest the time to revise the review article and the references. Lots of grammatical mistakes, typos, mismatch in Figure’s captions and text, undeleted comments and confusing sentences can be found. Although the contents of the article is interesting to the reader in this particular scientific community but the authors wasted their good efforts by submitting this unrevised version. I recommend its publication after proper revision.
1. Authors show many kinds of references and well introduce about the various techniques around this field.
On the other hand, it might be lack of some analytical point of view for these references. If the authors add more information about their purpose, their achievements, their contributions on the field and the summary of pros and cons, this would be much better. In terms of that, most of the references are introduced based on what they did in those papers, not based on what kinds of problem they can solve by this technique and why those techniques work well for solving the problem. I think concise summary of those would also be a great help for the readers. Also, the points of discussion which authors indicate in the beginning of a paragraph are not often discussed in the latter sentences or sometimes including contradiction, so I would recommend to modify those points as well.
2. I suggest the authors provide a paragraph about the types of different graphene materials and the advantages of using these materials. The comparison of graphene with other carbon nanomaterials such as carbon nanotubes should be described.
3. The reason for different voltage windows for different polymers should be included.
4. Some relevant papers should be included. For example, Materials Research Bulletin, Elsevier, 2017, 96, 395-404. Journal of Materials Chemistry A, RSC, 2015, 3, 22507-22541.
5. Some of the concrete examples I pointed out above are shown below, but similar kinds of issues are in many places in this manuscript. So I would recommend to modify similar points as well.
-The Line 96-97 “In general, CPs with a good specific capacitance and high conductivity could be considered very suitable for supercapacitor electrodes” is ambiguous and needs to be modified.
-Figure 3h should be put first as Figure 3 a rest of the images should be put later.
-In line 287 “a reasonably rectangular CV shape even at 400 mV s-1 (Figure 4e)” Figure 4e should be Figure 6 e.
-Figure 7 is wrongly addressed in text.
- Line 379 “The PEDOT/SDS had a leaf-like morphology (Figure 8a) and irregular block aggregates (Figure 8d)”, Figure 8d should be figure 8b.
-Line 380-383 reported that “On the other hand, PEDOT/SDS-GO films show a petal-shaped morphology, and this petal-shaped and porous composite film provided a large surface area, limited aggregation and enhanced ion and electron transport, during charge and discharge (Figure 8b).” However there is no charge-discharge data in same figure.
Author Response
Point 1: Authors show many kinds of references and well introduce about the various techniques around this field. On the other hand, it might be lack of some analytical point of view for these references. If the authors add more information about their purpose, their achievements, their contributions on the field and the summary of pros and cons, this would be much better. In terms of that, most of the references are introduced based on what they did in those papers, not based on what kinds of problem they can solve by this technique and why those techniques work well for solving the problem. I think concise summary of those would also be a great help for the readers. Also, the points of discussion which authors indicate in the beginning of a paragraph are not often discussed in the latter sentences or sometimes including contradiction, so I would recommend to modify those points as well.
Response: We thank this referee for the valuable comments. In this revision we have modified the context following the suggestions from this referee. For instance, when introducing others’ work, we added comments. In page 4, line 133, we added: “…, which is not high compared to the theoretical capacitance of PANI (~2000 F g-1).” Other similar changes are too many to be listed here.
At the end of each section, we have comments and discussions on others reports. After which, we give the section of “Conclusions and future perspectives”.
Point 2: I suggest the authors provide a paragraph about the types of different graphene materials and the advantages of using these materials.The comparison of graphene with other carbon nanomaterials such as carbon nanotubes should be described.
Response: In page 2, lines 56-61, we introduced the previous applications of CNTs in supercapacitors, and then introduced the work of graphene-based supercapacitors, written as “Over the past few years, many carbonaceous nanomaterials were used for supercapacitor electrodes. Single-walled carbon nanotube (SWCNT) is one of commonly used carbon nanomaterials, with a theoretical specific surface area of 1300 m2/g and also exhibit good specific capacitance as supercpacitor electrodes [23]. However, the limited surface area of SWCNT and high production cost restrict their applications for supercapacitor [24]. In comparison, graphene is a promising two-dimensional (2D) material for supercapacitors, with a theoretical specific surface area of 2630 m2/g [25].” Following this part we go into the introduction of graphene materials prepared by using different methods, such as CVD and the traditional Hummer’s method, and their properties.
Point 3: The reason for different voltage windows for different polymers should be included.
Response: The fundamental studies elucidating the different potential windows of conducting polymers are still absent in the literature. In our experiment, when tested using a three-electrode setup, graphene with small redox-active molecules adsorbed in the surface shows a voltage window from -0.2 V to 0.8 V in H2SO4 electrolyte. However, if the small molecules are electropolymerized, the voltage window can be extended, for e.g., from -0.6 V to 1.0 V. Further investigations are needed to answer this question.
Point 4: Some relevant papers should be included. For example, Materials Research Bulletin, Elsevier, 2017, 96, 395-404. Journal of Materials Chemistry A, RSC, 2015, 3, 22507-22541.
Response: The two papers are cited as Refs. 34 and124 in this revision.
Point 5: Some of the concrete examples I pointed out above are shown below, but similar kinds of issues are in many places in this manuscript. So I would recommend to modify similar points as well. -The Line 96-97 “In general, CPs with a good specific capacitance and high conductivity could be considered very suitable for supercapacitor electrodes” is ambiguous and needs to be modified.
-Figure 3h should be put first as Figure 3 a rest of the images should be put later.
-In line 287 “a reasonably rectangular CV shape even at 400 mV s-1 (Figure 4e)” Figure 4e should be Figure 6 e.
-Figure 7 is wrongly addressed in text.
- Line 379 “The PEDOT/SDS had a leaf-like morphology (Figure 8a) and irregular block aggregates (Figure 8d)”, Figure 8d should be figure 8b.
-Line 380-383 reported that “On the other hand, PEDOT/SDS-GO films show a petal-shaped morphology, and this petal-shaped and porous composite film provided a large surface area, limited aggregation and enhanced ion and electron transport, during charge and discharge (Figure 8b).” However there is no charge-discharge data in same figure.
Response: For the sentence, “In general, CPs with a good specific capacitance and high conductivity could be considered very suitable for supercapacitor electrodes”, it has been changed into “CPs with high theoretical specific capacitances and high charge mobilities are considered as good pseudocapacitive materials for building supercapacitors with higher energy densities, compared to the conventional carbon-based EDLCs.”.
Figure 8 has been updated by re-arranging the order of panels and adding two new panels to show the charge-discharge data.
Other typo problems have been corrected.
Reviewer 3 Report
This manuscript reviewed the “Recent supercapacitors based on graphene and redox-active organic materials”. The recent research progress in graphene/CP composites for the application of supercapacitors are summarized in detail. I can clearly recommend acceptance of this interesting review. However, I suggest some minor changes before final acceptance.
1. The conclusions can be improved, it is hard to understand the conclusion part.
2. Also, future perspectives are not clear, specific future perspectives are missing.
3. References should be re-checked carefully, issue numbers were added for some references only and end page numbers are missing for some references.
4. Page numbers are missing for some references.

Author Response
Point 1. The conclusions can be improved, it is hard to understand the conclusion part.
Response: In the conclusion, page 28, lines 788-794, we added “However, currently the available CPs for supercapacitors are limited to PANI, PPy, and PEDOT. Hence their drawbacks in terms of poor cycling stability and narrow voltage window in aqueous electrolyte are anticipated to be solved by the synthesis of new-type CPs. Building asymmetric supercapacitors by using two dissimilar CPs, one as the negative electrode and the other one as the positive electrode, can be an efficient way towards high performance supercapacitors with high specific capacitance and broad voltage window beyond the critical voltage of water splitting (~1.23 V).”
Point 2: Also, future perspectives are not clear, specific future perspectives are missing.
Response: We have modified the sections of ‘future perspectives’ by addressing some issues on how to improve the electrochemical performance of CP-based supercapacitors and giving some application examples of such devices.
Point 3: References should be re-checked carefully, issue numbers were added for some references only and end page numbers are missing for some references. Page numbers are missing for some references.
Response: The format of all references has been corrected.
Round 2
Reviewer 2 Report
The queries raised are addressed successfully and manuscript can be accepted in its current form.
Author Response
The reviewer's coments:The queries raised are addressed successfully and manuscript can be accepted in its current form.
Hence no response is needed.